# Recent Advances in the Application of Mesenchymal Stem Cell-Derived Exosomes for Cardiovascular and Neurodegenerative Disease Therapies

**DOI:** 10.3390/pharmaceutics14030618

**Published:** 2022-03-11

**Authors:** Zhimin Yang, Yanyu Li, Zihua Wang

**Affiliations:** 1Fujian Provincial Key Laboratory of Brain Aging and Neurodegenerative Diseases, School of Basic Medical Sciences, Fujian Medical University, Fuzhou 350122, China; yangzm2021@nanoctr.cn; 2Department of Radiology, State Key Laboratory of Complex Severe and Rare Diseases, Peking Union Medical College Hospital, Chinese Academy of Medical Sciences and Peking Union Medical College, Beijing 100730, China; liyanyu324@163.com

**Keywords:** mesenchymal stem cell, exosomes, cardiovascular diseases, neurodegenerative diseases, drug delivery

## Abstract

Exosomes are naturally occurring nanoscale vesicles that are released and received by almost all cells in the body. Exosomes can be transferred between cells and contain various molecular constitutes closely related to their origin and function, including proteins, lipids, and RNAs. The importance of exosomes in cellular communication makes them important vectors for delivering a variety of drugs throughout the body. Exosomes are ubiquitous in the circulatory system and can reach the site of injury or disease through a variety of biological barriers. Due to its unique structure and rich inclusions, it can be used for the diagnosis and treatment of diseases. Mesenchymal stem-cell-derived exosomes (MSCs-Exo) inherit the physiological functions of MSCs, including repairing and regenerating tissues, suppressing inflammatory responses, and regulating the body’s immunity; therefore, MSCs-Exo can be used as a natural drug delivery carrier with therapeutic effects, and has been increasingly used in the treatment of cardiovascular diseases and neurodegenerative diseases. Here, we summarize the research progress of MSCs-Exo as drug delivery vectors and their application for various drug deliveries, providing ideas and references for the study of MSCs-Exo in recent years.

## 1. Introduction

Accompanied by the aging population, the incidence of cardiovascular disease (CVD) [1] and central nervous system disorder/disease (CNSD) [2] has been rising globally. CVD has been diagnosed and treated by many advanced methods. However, it remains the most prevalent disease and the most common cause of death worldwide [3,4]. These conditions largely depend on the death of cardiomyocytes and subsequent abnormal tissue remodeling [5]. CNSD can be divided into several types of disorders. Neurodegenerative diseases are featured by accumulating misfolded proteins and resulting in a progressive decline in neuronal function, such as Alzheimer’s disease (AD) or Parkinson’s disease (PD). Infectious diseases are characterized by having explicit pathogens in the brain or spinal cord tissue, including encephalitis or poliomyelitis. Cerebrovascular diseases include stroke, transient ischemic attack (TIA), and cerebral infarction. Oncological diseases include glioma, meningioma, etc. [6]. As myocardial cells and neuronal cells are non-renewable, CVD and CNSD are commonly regarded as irreversible impairments, and the treatment of these diseases is greatly blocked. In the treatment of disease, MSCs-Exo can be used as a drug carrier to transfer different drugs to the disease center with high efficiency and precision. Although researchers have paid much attention to the early diagnosis and treatment of CVD/CNSD, there are still some challenges to overcome [7]. The emergence of stem cell therapy or exosomes therapy provides a new diagnosis/treatment tool. This review article aims to discuss the research progress based on MSCs/MSCs-Exo to treat CVD or CNSD, explore the potential mechanisms, and as a tool in nanomedicine.

## 2. Stem Cell Therapy for CVD/CNSD

A stem cell is a cell with infinite or immortal self-renewal capacity capable of producing at least one highly differentiated progeny cell. Stem cells have enormous potential in regenerative medicine by secreting many cytokines, proteins, growth factors, and extracellular vesicles for many non-effective therapy diseases. At the end of the last century, stem cell therapy [8] appeared and turned out to be an alternative biological treatment to traditional medical or surgical treatment [9,10]. At first, stem cell therapy was limitedly used for hematological malignant tumors. At the same time, with the evidence increased, experts found that pluripotent stem cells have the ability to differentiate into various cell types, migrate to multiple tissues, and secrete a variety of immunomodulation factors. Subsequently, stem cell therapy has become a “hot topic” for regenerative medicine and as a treatment for many diseases in basic research and preclinical or clinical studies [11,12]. According to the origins, stem cells can be divided into embryonic stem cells (ESCs), which are separated from embryos, and adult stem cells (ASCs) which originate from mature tissues [13]. In contrast with ESCs, ASCs, especially MSCs, have become the focus in medical research [14]. MSCs can self-renew and have multiple differentiation potentials, the same as ESCs. They are also easier to obtain because various tissues can donor them, including the umbilical cord, endometrial polyps, menstrual blood, bone marrow, and adipose tissue. Furthermore, MSCs have many advantages, such as the lower risk of stimulating immune responses or tumor development and minimal ethical issues to use these cells in the clinic [15]. Based on their advantages, MSCs have become the most potent candidate for medical research [16].

The treatment based on MSCs for CVD has shown many beneficial effects such as reducing fibrosis, stimulating angiogenesis, cardiomyogenesis [15]. Studies have shown that the multiple differentiation potentials of MSCs enable themselves to successfully differentiate into various cardiac cell types, such as cardiomyocytes, vascular endothelial cells, and vascular smooth muscle cells [17,18,19]. This feature allows MSCs to replace the damaged myocardium directly. In contrast, subsequent studies demonstrated that this differentiation capacity of MSCs was minimal. Additionally, researchers came up with a hypothesis that MSCs secrete molecules and microvesicles (including proteins, lipids, and nucleic acid) to regulate the microenvironment, reduce inflammatory reactions, and promote the repair of damaged tissue [20].

Currently, intravenous transplantation, myocardial injection, and endocardial injection are the top three methods for stem-cell-based cardiac therapy [21]. Hare et al. found that allogeneic MSCs and autologous MSCs delivered by transendocardial injection in patients can improve cardiovascular function and reduce the incidence of serious adverse events in ischemic cardiomyopathy (ICM) [22]. The same group intravenously transplanted allogeneic MSCs in patients with myocardial infarction (MI) and found that MSCs can increase left ventricular ejection fraction and lead to reverse remodeling, while the adverse event rates were similar to placebo [23]. Williams et al. demonstrated that in a swine MI model, human cardiac stem cells and bone marrow MSCs (bmMSCs) co-injection could reduce infarction size and restore cardiac function [24]. Premer et al. found that in dilated cardiomyopathy (DCM) patients, MSC secrete higher levels of SDF-1α which can modulate endothelial function [25]. Linthout et al. found that co-culture of MSCs with Coxsackievirus B3 (CVB3)-infected HL-1 cardiomyocytes resulted in a reduction in CVB3-induced HL-1 apoptosis, oxidative stress, and decreased the incidence of myocarditis [26].

MSCs transplantation in neurodegenerative disease animal models has improved various parameters, including increased survival rates, decreased tissue injury size, and improved cognitive capacity [27,28,29]. Additionally, in cerebrovascular diseases, some clinical trials have been performed [30,31]. Although the initial thought of MSCs therapy aimed to replace the repaired neurons, evidence suggests that MSCs promote postischemic neurological recovery by secreted neurotrophic factors instead of replacing disabled neurons. Studies have confirmed that MSCs facilitate neurological recovery and neo-angiogenesis [32,33,34] through the secretion of neuroprotection and angiogenesis regulatory factors. These previous studies illustrated that the secretion of MSCs might produce neuroprotection by improving the growth of neo-angiogenesis.

With the evidence accumulating, subsequent studies demonstrate that MSCs directly do not have a particular treatment function in diseases [35]. Experts even found that when systemically administered MSCs, only a tiny amount of exogenous MSCs can localize to injured tissue, and the right localized MSCs are also able be rapidly cleared. In addition, the homing mechanism of MSCs is not fully clear also has influences on the homing capacity of MSCs, which may be resulted in a significant reduction in the therapeutic effect. Even worse, the administration of MSCs may elicit adverse effects [36], such as immune reactions [37,38,39], graft versus host disease [40], secondary infection [41], and canceration [42,43]. Coincidently, experts found that the supernatant of MSC cultures has the parallel function to improve disease conditions. This inspires the idea that the secretions of MSCs might have a similar effect for treating disease.

## 3. Exosomes as a Therapeutic Tool

In the 1980s, extracellular vesicles (EVs), which can be secreted by almost all cells, were first discovered by P. Stahl [44] and R. Johnstone [45]. The biogenesis, secretion, and action mode of EV are illustrated in Figure 1. EVs are membrane-contained small vesicles, secreted by all types of prokaryote and eukaryote cells. The main category is apoptotic bodies, shedding micro vesicles and exosomes. In both physiological and pathological conditions, they can participates in cellular communication and convey specific information in origin and target cells; in physiological conditions, they can regulate homeostasis; or in pathological conditions, they can have bad effects, increasing tumorigenesis and metastasis, inflammation, and immune system activation [46]. Exosomes as a type of EV have Characteristics of the EVs and also have their own unique characteristics. They are membranous vesicles containing proteins, lipids, mRNAs, microRNAs, and other non-coding RNAs. Exosomes are cargos enclosed by a lipid bilayer membrane which contain cholesterol, phosphatidylserine, and sphingomyelin and ceramide and express a special subset of surface proteins such as membrane transport/fusion proteins, heat shock proteins (HSPs), and tetraspanins (CD9, CD63, CD81). In addition to electron microscopy used as a golden standard for identification of exosomes, the presence of CD9, CD63, and CD81 in exosomes helps facilitate their detection. Of note, exosomes may also express certain surface proteins, which define a very specific profile of exosomes [47]. According to body size, biological characteristics, and formation processes, EVs are mainly separated into exosomes, microvesicles, and apoptotic bodies [48]. At first, EVs were initially considered to be receptacles of cell waste biomaterial with no particular function. Later, researchers found that the EVs can achieve their biological activity by paracrine action of cells. Hence, increasing articles have emerged about the separation, identification, and function of EVs. Different methods have been used to isolate EVs from cells, including ultracentrifugation [49,50], immunological separation [51], ultrafiltration [52], size exclusion chromatography [52], polymer-based precipitation separation [53], magnetic separation [54], acoustic fluid separation, acoustic fluid separation [55], deterministic lateral displacement separation [56]. During the isolation technique, the most commonly used is differential centrifugation.

Exosomes, the prominent EVs, typically have a cup-shaped structure under the electron microscope with a size of 30–150 nm. Exosomes are smaller than other EVs, and their size distribution is more uniform. As large and aggregated vesicles (>200 nm) can be trapped in the sinusoids or can be swallowed by macrophages, after systemic administration, smaller vesicles, like exosomes, can cross the endothelial barrier [57]. Exosomes are derived from their originating cell with bilayer membranes [58]. The membrane component of exosomes has a higher cholesterol content [59] and has a lower phosphatidylcholine level than donor-cell membranes [60]. The characters make them less susceptible to the permeation of small solutes and are more stable, and allow them to fuse with the receptor cells and release their contents inside [61]. Aside from their lipid composition, the surfaces of exosomes contain proteins and sugars, which makes exosomes change charge and maintenance of membrane structure, and can be the biomarker for exosome identification [62]. The contents of exosomes are lipids, proteins, and nucleic acids, such as microRNAs, etc. [63,64], which play an essential role in intercellular communication and shows the possibility for using exosomes as a tool to treat disease or as biomarkers for early diseases diagnosis [36]. Although natural MSCs-Exo showed no significant difference in efficacy compared with MSCs transplantation, MSCs-Exo is more feasible for clinical application in the treatment of central nervous system injury due to its higher safety and stronger plasticity compared with MSCs. Currently, numerous studies have shown that exosomes are therapeutic in a variety of conditions, including neurodegenerative diseases [65], cardiovascular [66], and cerebrovascular diseases [67], and so on [68]. MSCs-derived exosomes (MSCs-Exo) not only have the characteristics of MSCs but the advantages of EVs (Table 1). In contrast with MSCs, they use MSCs-Exo as cell-free therapeutics that offer several benefits, such as high stability, accessible storage, and low immunogenicity [69,70,71,72].

## 4. Exosomes Therapy for Cardiovascular Disease

The recent developments in exosomes therapy for CVD have been summarized in Table 1. It has been confirmed that MSCs transplantation helps recover the injured cardiomyocytes and preserves heart function in vivo and in vitro [15,18,82,83,84]. Focusing on these features, MSCs-Exo is also under-researched in many basic medical or preclinical studies [85]. In 2010, the first study of MSCs-Exo was used for myocardial ischemia/reperfusion injury. Researchers proposed that MSCs-Exo contributes to myocardial repair [86]; however, the underlying mechanism is not entirely clear. Soon after that, several studies have illustrated the similar efficacy of MSCs-Exo for CVD treatment, and researchers tend to seek out the potential mechanism. Zhao and his colleagues found that MSCs-Exo attenuates myocardial ischemia/reperfusion injury via modifying the polarization of M1 macrophages to M2 macrophages. The possible signal system may be miR-182, a potent candidate mediator of macrophage polarization, and Toll-like receptor 4 (TLR4) as a downstream target [73]. Zhu et al. used hypoxia-conditioned MSCs-Exo (Hypo-MSCs-Exo) to administer to mice with a permanent condition of MI and found that the content of Hypo-MSCs-Exo facilitates ischemic cardiac repair by ameliorating cardiomyocyte apoptosis [74]. Another study by Lai et al. [86] showed that the infarct size was remarkably reduced by administrating MSCs-Exo in mice models of myocardial ischemia/reperfusion injury. Huang and his colleagues administered bmMSCs-derived exosomes (bmMSCs-Exo) to a rat model of acute MI (AMI) and demonstrated that MSCs-Exo could reduce scar size and restore heart function [75]. Sun et al. have shown that exosomes derived from using lentivirus containing HIF-1α overexpressing vector infected MSCs can rescue the impaired cardiac tissue by promoting neovessel formation and inhibiting fibrosis [76]. Bian et al. [77] administered MSCs-EVs to a rat model of AMI, reduced the scar size and preserved the diastolic and systolic function. Hirai and his group found that intracoronary injection of cardiosphere-derived exosomal microRNAs is safe and improves cardiac function in a swine model of dilated cardiomyopathy (DCM) [78]. In a study, Yu and his group found that this protective effect of MSCs-Exo was partly due to the content of exosomes, such as miRNA [79].

In recent years, many studies have also been attempting to identify various diseases by the early prognosis value of MSCs-Exo. Recent studies have suggested that plasma levels of exosome-related miRNAs could serve as new biomarkers in disease diagnosis [87]. These findings suggest that exosomes may also have some value in disease diagnosis.

## 5. Exosomes Therapy in Central Nervous System Disease

Neurodegenerative diseases are chronic, latent, progressive disorders that occur in the central nervous system (CNS). They are characterized by the loss of neuronal structure and function [88] lead to Information transmission disorder for neuron-to-neuron. Although great progress has been made in neuroscience research, there are still huge shortcomings. The main and most important is the lack of thorough understanding of its specific pathological mechanism. Moreover, the diseases of the nervous system are more complicated and multifactorial, which makes it more difficult to study [6]. Due to the extremely weak self-regeneration of neurons, the dysfunction after central nervous system injury often accompanies patients for life, so it has become a difficult problem to be broken through in clinical treatment. In the treatment of central nervous system injury, MSCs-Exo has been proved to have definite therapeutic effects of promoting neurovascular regeneration, regulating inflammatory environment and repairing nerve myelin sheath. Many researchers have confirmed that stem-cell-derived exosomes have neuroprotection and neurotrophy effects [36,89] (Figure 2). Furthermore, the small size of MSCs-Exo makes itself cross the blood–brain barrier, reaching the brain or spinal cord tissue [36]. The recent developments in exosomes therapy for CNSD have been summarized in Table 2. Recent studies have found that independent MSCs-Exo can promote the regeneration of endogenous neurons and vascular endothelial cells in central nervous system injury lesions [90]. In animal models, MSCs-Exo treatment can improve the function defect caused by central nervous system injury, and has obvious therapeutic effect [91].

Stroke is one of the leading causes of death [96]. Some cell-based therapies have been deliberated to promote stroke treatment in preclinical and clinical trials [97], and the secretion of stem cells, such as exosomes, has also shown to be promising in various preclinical models of stroke recovery [98]. Xin et al. found that in rat models of stoke, administration of MSCs-Exo has been shown to enhance neurogenesis and angiogenesis and improve brain function recovery [95]. In amyotrophic lateral sclerosis (ALS), related studies have found that misfolded SOD1 protein can be transferred from cell to cell through exosomes dependence and exosomes independence, make progating diaeases [99]. In addition, another study indicated that TDP-43 was also associated with exosomes. Exosomes are an important pathway for THE transfer of TDP-43 aggregates [100]. Both SOD1 protein and TDP-43 protein are important pathological features of ALS. Exosomes also act through anti-apoptosis and anti-necrosis mechanisms (activating cell survival PI3K-B-cell lymphoma-2 (Bcl-2) pathway). Additionally, endogenous neuronal survival factors play an important role in the treatment of ALS by enhancing the receptor cells [101]. Riazifar and their colleagues assessed the effect of MSCs-Exo in treating multiple sclerosis and found that intravenous administration of MSCs-Exo can decrease neuroinflammation and reduce demyelination [65]. Additionally, in multiple sclerosis it often presents as chronic inflammation. MSCs-exo can also play an important role in this regard. It can regulate the activation of microglia by and by inhibiting the release of pro-inflammatory cytokines, greatly reducing the amount in the plasma to reduce inflammatory infiltration. A study [102] used IFNγ-stimulated dendritic cell cultures to release exosomes that increased myelin levels and reduced oxidative stress and promote myelin reformation after demyelination. Chen and their partners administrated human adipose mesenchymal stem cell (HaMSC)-derived exosomes (HaMSC-Exo) into a weight-drop-induced traumatic brain injury (TBI) rat model and found that HaMSC-Exo promoted functional recovery, suppressed neuroinflammation, reduced neuronal apoptosis, and increased neurogenesis in TBI rats [80]. In a study to test the influence of MSCs-Exo in a large animal model of TBI, experts used MSCs-Exo to administrate female Yorkshire swine after TBI. They found that exosome-treated animals had significantly attenuated brain swelling and smaller lesion size, decreased blood-based cerebral biomarkers levels, and improved blood–brain barrier (BBB) integrity [81]. The inflammatory response activated after nerve injury may cause a secondary attack on the lesion. However, MSCs-Exo significantly prevented the pro-inflammatory cytokine release while promoting an M1 to M2 phenotype polarization in microglia and thereby reducing inflammatory damage. de Godoy, Mariana A. et al. have shown that MSC-EVS can reduce the expression of ROS related fluorescence signal in AD hippocampus neurons in vitro and protect neurons from Aβ protein-induced oxidative damage [103]. Recent studies have found that MSCs-Exo can significantly reduce the accumulation of β-amyloid (Aβ) protein in neurons, which confirming the therapeutic effect of MSC-EVS on the alleviation of pathological changes in Alzheimer’s disease [104]. In AD mice model, experts found that intracerebroventricularly injected BM-MSCs can improve cognitive impairment by ameliorating astrocytic inflammation as well as synaptogenesis [93]. In another study, MSCs-Exo was injected into APP/PS1 mice, and after a period of treatment, the ability of spatial learning and memory was significantly improved. The symptoms associated with AD were significantly improved. It was confirmed that the activation of SphK/S1P signaling pathway could reduce Aβ deposition and promote the recovery of cognitive function in AD mice [105]. As neurodegenerative diseases are characterized by the intracellular or extracellular aggregation of misfolded proteins [106], some experts intend to find early content changing of exosomes as biomarkers for AD/PD diagnosis. Yang et al. found that the serum exosomes-derived microRNA, miR-135a, -193b, and -384, are potential biomarkers for early AD diagnosis. In addition, MSC-EVs also has a regulatory effect on the microglial immune activated by Aβ, which can improve the neuronal survival in AD brain. Research found that MSC-EVS could inhibit microglia polarization to pro-inflammatory M1 subtype and increase the number of anti-inflammatory M2 subtype microglia in AD transgenic mice, and upregulate the expression of anti-inflammatory progenitor TGF-β and IL-10 in brain tissues [107]. This immunomodulatory effect is also involved in the protective effect of MSC-EVs on AD neurons. Additionally, researchers discovered that the exosomes from AD patients might become toxicity vesicles containing toxic amyloid-beta protein [108]. That also illustrates that exosomes are closely related to the occurrence of CNSD. Wang et al. development of exosome as a carrier for curcumin prevents neuronal death in vitro and in vivo to alleviate AD symptoms. This study provides potential clinical evidence for exosome-based drug delivery in the treatment of AD [109]. Meckes Jr et al. found that 5 × FAD mice received hMSCs-Exo treatment can slow down AD pathogenesis and ameliorate inflammatory marker glial fibrillary acidic protein (GFAP) in a preclinical mouse model [92]. However, the efficacy of MSCs-Exo demonstrated only from the perspective of Aβ protein may require further validation in future clinical trials. Exosome-associated miR-137 has been found to be upregulated in neurons in PD, where it plays a vital role in neuronal oxidative stress induction. MiR-137 directly targets oxidation resistance-1 (OXR1) to negatively regulate its expression, thereby inducing oxidative stress. The levels of miRNAs have also been investigated in some PD models, such as in a manganese model where 12 miRNAs were significantly increased in exosomes; these miRNAs were shown to regulate key PD pathogenesis pathways including autophagy, inflammation and protein aggregation [110]. Another group reported that exosome delivery of hydrophobically modified siRNA to the brain efficiently targeted mHtt mRNA in a Huntington’s disease model, which is encouraging for the potential use of siRNAs to target α-syn in PD [111]. MSC-derived exosomes proved effective at rescuing dopaminergic neurons in the 6-OHDA mouse model of PD, and they can also carry miRNAs and interact with neuronal cells to reduce neuroinflammation and promote neurogenesis in mouse PD models [112]. In another study, also using a 6-OHDA mouse MODEL of PD, treatment with SHED-derived exosomes was carried out. The expression level of TH in striatum and substantia nigra was decreased, demonstrating the potential of exosomes in PD treatment [113]. Whilst further investigations and clinical trials are required to confirm the benefits of therapeutic application of exosomes in PD, mounting evidence supports that the separation of exosomes from various cell types and their modification to target specific brain regions may hold therapeutic benefits for PD, among other disorders [114].

With the continuous development of MSCs-Exo research, a large number of research achievements have been made on the therapeutic effect and potential mechanism of MSCs-Exo in a variety of CVD and CNSD, showing great potential for disease treatment. However, at present, the treatment-related research of MSCs-Exo is still in the early stage, and there are many research gaps and problems to be explored and solved [115,116]. First, exosome bioactivity must be detected precisely. Only when exosomes have biological activity can they show function for treating or diagnosing various diseases. Second, exosomes need to be monitored in vivo for tracking biodistribution and targeting. Third, there need standardizations for exosomes composition, using dose, production, etc. In addition, the question about how to modulate the bioactivity of exosomes is also required to be considered.

## 6. Exosomes: Nano-Drug Delivery Vehicles

Although exosomes secreted by all types of cells (not just those secreted by mesenchymal stem cells) can play a large or small role in the treatment of neurodegenerative diseases. However, the ability to target exosomes in vivo is poor, and few exosomes are found to be delivered to the brain after intravenous injection [117]. Therefore, it is difficult to apply exosomes directly to the treatment of neurodegenerative diseases, but it has a broader prospect to use exosomes as vectors for treatment. Exosomes can escape phagocytosis and achieve long-term circulation by the ‘do not eat me’ signal for high levels of CD47 on exosomes and trigger CD47-SIRPα interaction that induce immune evasion. In addition, as endogenous cellular compartments, exosomes possess a wide range of cellular adhesion molecules facilitating their penetration through biological barriers [118]. In the treatment and treatment of diseases, exosomes can be used as drug carriers to transfer different drugs to the site of disease with high efficiency and accuracy [119]. The first study of using exosomes as a drug delivery system was reported in 2011, and after that, the idea has gained increasing attention [120]. As we all know, to have a good therapeutic effect on the treatment of neurological diseases, the ability to cross the blood–brain barrier is required, and exosomes themselves have this excellent ability; using this property alone as a drug carrier to help drugs cross the blood–brain barrier is of great value. Researchers loaded Catalase into exosomes and delivered it intranasally to PD mice; consequently, a large number of exosomes were detected in the mouse brains [121]. Yang et al. revealed this phenomenon more directly by using transgenic zebrafish models. Exosomes were coated with fluorescently labeled anticancer drugs to evaluate the distribution of anticancer drugs in the brain. Images of zebrafish show that the anticancer drug enters the brain in large quantities when it is unable to cross the blood–brain barrier [122].

In comparison with other engineered nanoparticles, exosomes, as natural nano-scale particles, have various advantages (the excellent biocompatibility, adjustable targeting efficiency, and stability to transfer biological materials to recipient cells) as drug delivery carriers and have been regarded as potential nano-drug delivery vehicles [14,123]. In the case of conventional nanoparticles, nonspecific interactions between nanoparticles (NPs) and plasma proteins occur immediately after the NPs enter the bloodstream. This leads to the formation of protein crowns, which then replace the original NPs, NPs is immediately recognized by the body’s immune system and eliminated quickly. Moreover, NPs is toxic and has potential safety risks [124]. Exosomes are superior to synthetic lipids in drug delivery vectors. Exosomes can be derived from the organism itself; therefore, their immunogenicity is low, and they are well tolerated. Even exosomes of alien origin have great safety in treating diseases. Exosomes can also cross the blood–brain barrier into the blood circulation in the brain. Exosomes have intrinsic homing properties and can be artificially modified to express specific molecules or improve their targeting ability. Additionally, the biomembrane of MSCs-Exo can be modified to exploit the engineered exosomes via targeted ligands and developed into advanced drug delivery systems that might have a dramatic impact on the future of disease therapy. The construction of an effective exosome drug delivery system requires appropriate modification and modification of its different components to achieve the desired effect. The functionalization of exosomes can prolong the circulation time of exosomes, enhance the delivery efficiency of exosomes in cytoplasm and promote the targeting of exosomes.

At present, the commonly used exosome drug loading methods are divided into direct loading and indirect loading. Direct loading is the direct loading of drugs into purified exosomes through co-incubation, electroporation and transfection. Indirect loading is mainly through co-incubation or chemical transfection of drug and exosome donor cells to produce drug-loaded exosomes. However, there is a lack of horizontal comparison of different loading methods in the current research [119]. There are two critical approaches to incorporate therapeutic drugs into exosomes: active or passive loading/encapsulation [125]. Passive drug loading methods are relatively simple by incubating drugs with exosomes or incubating drugs with donor cells [126]. Such as proteins or peptides, which have great potential in treating neurological diseases. However, it is limited by its inability to cross the blood–brain barrier or its rapid degradation, preventing it from doing its full function. These two problems can be solved by loading them with exosomes. Haney et al. encapsulated soluble lysosomal enzyme tripeptidyl peptidase-1 (TPP1) into small exosome and intrabitoneally injected TPP1-EVs into neuronal paraffin lipobrown (LINCL) mice. It can reduce neuroinflammation and astrocyte proliferation, accumulate large amounts of EV carriers in the brain and prolong life [127]. The features of the active drug loading method are that it can enhance drug loading efficiency and load large molecules [125]. Because MSCs-Exo is easily absorbed by cells and has stronger tumor susceptibility after modification. Zhu et al. developed a c(RGDyK)-modified and paclitaxel (PTX)-loaded ESC-exos significantly improves the curative effects of PTX in glioblastoma via enhanced targeting, reducing the occurrence of dose-dependent side effects of chemotherapy drugs [128].

Exosomes can also load small molecules and RNA, Alvarez-Erviti et al. developed an exosomes-endogenous nano-vesicles that transport siRNA to the brain of AD mice to strong knockdown of Alzheimer’s disease-related gene BACE1 [129]. Jahangard et al. loaded MiR-29 in MSC-EXOs to downregulated BACE1 and BIM in the hippocampus that reduced the pathological effects of Aβ in a rat model of AD [130]. Dopamine is important in the treatment of Parkinson’s disease, but dopamine cannot cross the blood–brain barrier, intravenous injection is prone to complications. After exosome encapsulation, dopamine was significantly transported to the striatum and substantia nigra to improve Parkinson’s symptoms [131]. The presence of a-synuclein aggregates in Louis subtitle is an important pathological characterization of PD. A-synuclein-siRNA-EVS can reduce the expression of a-synuclein mRNA, thus reducing its protein level [6]. Chronic inflammatory injury plays an important role in the course of neurodegenerative diseases. In a study of septic shock, compounds formed by curcumin inclusion in exosomes were found to bind to activated monocytes. Induced monocyte apoptosis and reduced oxidative stress injury, the formation of exosome curcumin complex also increased curcumin stability in vitro and bioavailability in vivo [132]. Curcumin has recently been shown to have great potential in the treatment of AD [133]. The combination of curcumin and exosome in the treatment of AD will have a bright future. This study proved that engineered exosomes had a specific protective effect on amyloid pathogenesis. Ma et al. used electroporation to load miR-132 into MSC-Exo to promote the formation of new blood vessels in the infarcted area and the recovery of heart function in the myocardial infarction mouse model [134].

## 7. Conclusions and Perspectives

In summary, MSCs-Exo, as a treatment for diseases free of stem cells, can not only guarantee the efficacy of diseases, but also has better safety and convenience compared with stem cell therapy, and will certainly play a huge role in the future clinical treatment of diseases. Exosome as the carrier of therapy because can be applied to a variety of diseases, to achieve the effect that conventional therapy cannot achieve, more and more researchers are favored, but also attract more capital investment, its prospects are unlimited. As biologically active nano-vesicles, MSC-exosomes have shown many advantages in disease treatment, such as cardiovascular disease, neurodegenerative disease, tumors, and regenerative medicine. Currently, an accumulating amount of evidence has been showing that MSCs-Exo has disease treating potential and can successfully apply for the therapy of several kinds of diseases model [135,136]. A few clinical trials are currently on-going but there are still challenges to overcome for further clinical translation such as the scale-up of the production, the lack of standardization for isolation and characterization methods and the low encapsulation efficiency. In contrast with MSCs, evidence suggests that MSCs-Exo promotes angiogenesis, restrains inflammatory effects, decreases immunogenicity, and reduces tumors production, which enable exosomes more possible to apply to the area of precision medicine.

As a therapeutic tool, compared with standard delivery methods, MSCs-Exo holds great therapeutic promise, but still faces many challenges. Due to the size and complexity of MSCs-Exo, technological limits exosomes are not only facing technical challenges, but also challenges in clinical practice, such as large-scale pharmaceutical production and production costs. Previously, scientists used ultracentrifugation to purify exosomes, which was a labor-intensive and time-consuming process that could not be used for large-scale production. To find new solutions in future research and develop a simple purification method with very low cost and safety. It can produce enough exosomes for clinical use. This is the focus of future research. Several primary issues need to be solved [94]. We should establish clinical standards of the isolation, quantification, and characterization of exosomes (Figure 3) [137]. This makes it easier to conduct clinical studies and translational applications. The therapeutic effects of MSCs-Exo may vary in different diseases. It is essential to eliminate unknown interference from the contents of exosomes. Furthermore, several critical technological issues have not yet been resolved, such as side effects of exosomes as drugs, optimal dosage, and route of administration. Consequently, future studies should be implemented to solve these problems and exploit more prospected engineered exosomes.

## Figures and Tables

**Figure 1 pharmaceutics-14-00618-f001:**
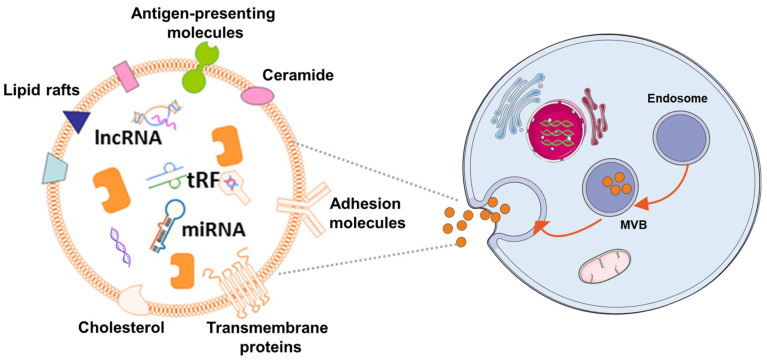
The mechanism of EV biogenes. Exosomes are released by cells of hematopoietic and non-hematopoietic origin. The EV production process can be summarized as follows: (1) the cytoplasmic membrane inward budding forms intracellular vesicles, which are termed as endosomes; (2) intracellular vesicles further develop to form multivesicular bodies (MVBs); (3) MVBs fuse with the cytoplasmic membrane to release exosomes, which can be incorporated into recipient cells through pinocytosis/phagocytosis or influence recipient cell signaling via ligand-receptor interaction.

**Figure 2 pharmaceutics-14-00618-f002:**
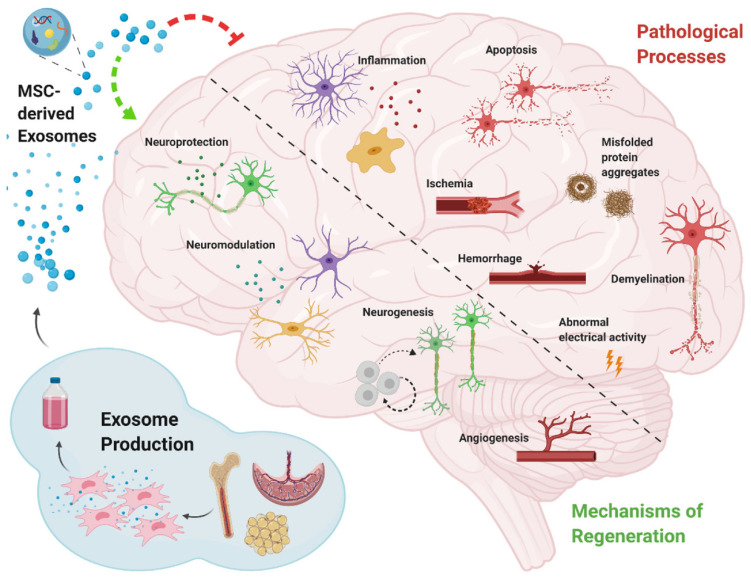
Schematic diagram of key mechanisms of mesenchymal stem cell-derived exosomes for the treatment of neurodegenerative diseases [36].

**Figure 3 pharmaceutics-14-00618-f003:**
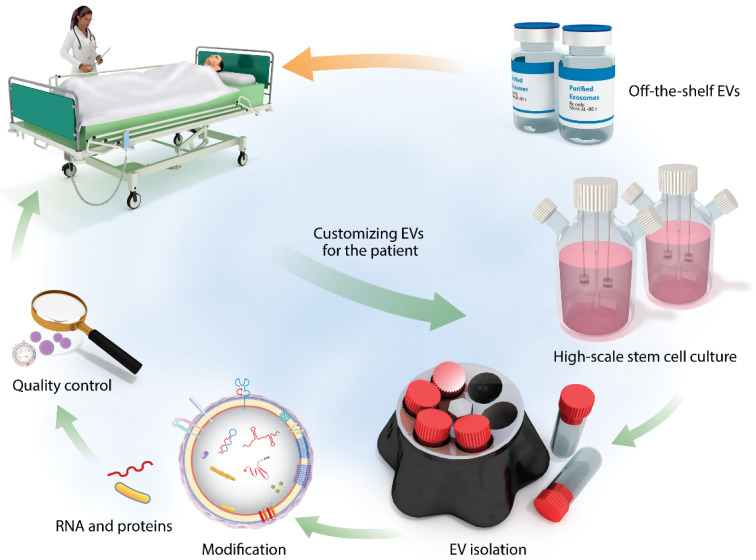
The process of EVs designed, manufactured, and quality controlled beforehand for clinical applications [137].

**Table 1 pharmaceutics-14-00618-t001:** Example of MSCs-Exo against cardiovascular diseases.

Human/Animal Model	Disease/Disorder	Cell/Cell Source	Administration Methord	Biological/Medical Improvement	Reference
patients	ICM	autologous and allogeneic hbmMSCs	transendocardial injection	reduce the incidence of serious adverse events	[22]
patients	AMI	allogeneic hbmMSCs	intravenous injection	increase left ventricular ejection fraction	[23]
Yorkshire swine	MI	hMSCs and hCSCs	myocardial injection of infarct border	reduce scar size restore diastolic and systolic function	[24]
patients	DCM	autologous and allogeneic MSCs	transendocardial injection	improve endothelial function	[25]
mice	myocarditis	hbmMSCs	Inject but not mention the root	improve murine acute CVB3-induced myocarditis	[26]
mice	myocardial ischaemia/reperfusion injury	MSC-Exo	myocardial injection of peri-infarct region	attenuate myocardial ischaemia/reperfusion injury	[73]
mice	MI	Hypo-Exo	intramyocardial injection	facilitate ischemic cardiac repair	[74]
rat	AMI	MSC-Exo	intramyocardial injection	reduce scar size	[75]
rat	MI	exosomes derived from HIF-1α-modified MSCs.	Not mentioned	promote neovessel formation and inhibite fibrosis	[76]
rat	AMI	hbmMSCs-Exo	intramyocardial injection	enhance blood flow recovery; preserve cardiac systolic and diastolic performance	[77]
swine	DCM	CDCs-Exo	intracoronary injection	Improve cardiac function and reduce myocardial fibrosis	[78]
rat	MI	MSCGATA-4-Exo	intramyocardial injection	Restore cardiac contractile function and reduce infarct size	[79]
rat/swine	TBI	haMSCs-Exo	intracerebroventricular injection	promote functional recovery	[80,81]

ICM = ischemic cardiomyopathy; EAE = Encephalomyelitis; MI = myocardial infarction; AMI = myocardial infarction acute; DCM = dilated cardiomyopathy; TBI = traumatic brain injury; AD = Alzheimer’s disease; hMSCs = human MSCs; hCSCs = human c-kit(+) cardiac stem cells (CSCs); hbmMSCs = human bone marrow MSCs; SDF-1α = stromal-derived factor-1α; EPC-CFU = endothelial progenitor cell-colony forming units; Hypo-Exo = hypoxia-conditioned bmMSC-Exo; CDCs-Exo = exosomes derived from cardiosphere-derived cells; MSCGATA-4 = mesenchymal stem cells (MSC) overexpressing GATA-4; haMSCs = human adipose mesenchymal stem cell.

**Table 2 pharmaceutics-14-00618-t002:** Example of MSCs-Exo against neurodegenerative diseases.

Human/Animal Model	Disease/Disorder	Cell/Cell Source	Administration Methord	Biological/Medical Improvement	Reference
mice	AD	MSCs-Exo	intranasally	attenuate amyloid beta (Aβ) and GFAP levels	[92]
mice	AD	bmMSCs-Exo	intracerebroventricular injection	improve cognitive impairment	[93]
Patients (phase 1/2)	AD	Allogenic adipose MSC-derived exosomes	nasal drip	Safety and Efficacy Evaluation of MSC-Exosomes in AD Patients	[94]
rat	stroke	bmMSCs-Exo	intravenous injection	improve functional recovery and enhance neurite remodeling, neurogenesis, and angiogenesis	[95]
patients	stroke	autologous bmMSCs	intravenous injection	reduce death rate	[30]
mice	EAE	MSCs-Exo	intravenous injection	decrease neuroinflammation and reduce demyelination	[65]

## Data Availability

Not applicable.

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
