# Peer review of "Recent Advances in the Application of Mesenchymal Stem Cell-Derived Exosomes for Cardiovascular and Neurodegenerative Disease Therapies"

_pharmaceutics, 2022, doi:10.3390/pharmaceutics14030618_

Round 1

Reviewer 1 Report

The manuscript by Yang et al. results interesting for the scientific community; it is accurate, well-organized and written.

Authors summarized, with appropriate references, the studies on the use of Stem Cells and in particular of Exosomes derived from Stem Cells for the therapies of cardiovascular and neurodegenerative diseases. In fact, MSC-Exo with the physiological functions of MSCs can be used as natural vectors for the administration of drugs with therapeutic effects. Furthermore, their use is easier for clinical application than MSCs due to their greater safety, plasticity and their low immunogenicity.

Although the manuscript is suitable for the publication in Pharmaceutics Journal, I think there are some issues that need to be addressed:

  1. In the section titled "Exosomes as a Therapeutic Tool", the authors described extracellular vesicles (EVs) and then Exosomes as a type of EVs. I think, in the paragraph, the authors should first describe EV and their general properties and then the exosomes. The description is not always linear.
  2. In addition along the text, in this specific paragraph and in figure 1 legend the description of exosomes resulted redundant: for example “EVs generally include active cargos such as proteins, lipids and nucleic acids.”
  3. I think that the authors should divide table 1 in two table: one refers to MsCs -Exo in cardiovascular diseases and on refers to MsCs -Exo in neurodegenerative diseases. In this way, readers are helped in understanding and can easily find the reference to a specific study.
  4. It is unclear whether Figure 1 refers to EVs or just to exosomes. Furthermore, the first part of the legend text should be reworded, because it is redundant.
  5. I think that the tables and figures should be insert near their first citation. All the abbreviations should be explain as footnotes in the table.
  6. Abbreviations must always be detailed the first time they appear in the text.

Author Response

  1. Comments and Suggestions for Authors

The manuscript by Yang et al. results interesting for the scientific community; it is accurate, well-organized and written.

Authors summarized, with appropriate references, the studies on the use of Stem Cells and in particular of Exosomes derived from Stem Cells for the therapies of cardiovascular and neurodegenerative diseases. In fact, MSC-Exo with the physiological functions of MSCs can be used as natural vectors for the administration of drugs with therapeutic effects. Furthermore, their use is easier for clinical application than MSCs due to their greater safety, plasticity and their low immunogenicity.

Although the manuscript is suitable for the publication in Pharmaceutics Journal, I think there are some issues that need to be addressed:

  1. In the section titled "Exosomes as a Therapeutic Tool", the authors described extracellular vesicles (EVs) and then Exosomes as a type of EVs. I think, in the paragraph, the authors should first describe EV and their general properties and then the exosomes. The description is not always linear.

Response: Thanks a for the reviewer's constructive comments and suggestion. According to your professional suggestion, the questions have all been carefully replied and the corresponding section has been extensively revised. All the revisions in the manuscript have been highlighted in red color and the grammatical and spelling problems have been double-checked. Please checked the updated version of manuscript.

2. In addition along the text, in this specific paragraph and in figure 1 legend the description of exosomes resulted redundant: for example “EVs generally include active cargos such as proteins, lipids and nucleic acids.”

Response:Thanks a for the reviewer's comments. According to your professional suggestion, the questions have all been carefully replied and the corresponding section has been extensively revised. Please checked the updated version of manuscript.

3. I think that the authors should divide table 1 in two table: one refers to MsCs -Exo in cardiovascular diseases and on refers to MsCs -Exo in neurodegenerative diseases. In this way, readers are helped in understanding and can easily find the reference to a specific study.

Response: According to reviewer suggestion, the questions have all been carefully replied and the corresponding section has been extensively revised. Please checked the updated version of manuscript.

4. It is unclear whether Figure 1 refers to EVs or just to exosomes. Furthermore, the first part of the legend text should be reworded, because it is redundant.

Response: According to reviewer suggestion, the corresponding section has been extensively revised. Please checked the updated version of manuscript.

5. I think that the tables and figures should be insert near their first citation. All the abbreviations should be explain as footnotes in the table.

Response: According to reviewer suggestion, the corresponding section has been extensively revised. Please checked the updated version of manuscript.

6. Abbreviations must always be detailed the first time they appear in the text.

Response: According to reviewer suggestion, the corresponding section has been extensively revised. Please checked the updated version of manuscript.

  1. Comments and Suggestions for Authors

The authors carry out this review work that aims to discuss the progress of research based on MSCs/Exo-MSCs to treat CVD or CNSD, explore the potential mechanisms and as a tool in nanomedicine. 

There is quite a bit of controversy about the nomenclature of extracellular vesicles depending on their formation, size, molecules they contain, etc. (Minimal information for studies of extracellular vesicles 2018 (MISEV2018): a position statement of the International Society for Extracellular Vesicles and update of the MISEV2014 guidelines. J Extracell Vesicles. 2018 Nov 23;7(1):1535750. doi: 10.1080/20013078. 2018.1535750. PMID: 30637094; PMCID: PMC6322352). However, the authors indicate in their work that they specifically reviewed exosome therapies. Why have they focused on that type of extracellular vesicle?

Response: Thanks a for the reviewer's constructive comments and suggestion. Several studies have reported that mesenchymal stem cells (MSCs)-derived exosomes have functions similar to those of MSCs, such as repairing tissue damage, suppressing inflammatory responses, and modulating the immune system. Compared with cells, exosomes are more stable and reservable, have no risk of aneuploidy, a lower possibility of immune rejection following in vivo allogeneic administration, and may provide an alternative therapy for various diseases.

Reviewer 2 Report

The authors carry out this review work that aims to discuss the progress of research based on MSCs/Exo-MSCs to treat CVD or CNSD, explore the potential mechanisms and as a tool in nanomedicine. 

There is quite a bit of controversy about the nomenclature of extracellular vesicles depending on their formation, size, molecules they contain, etc.        (Minimal information for studies of extracellular vesicles 2018 (MISEV2018): a position statement of the International Society for Extracellular Vesicles and update of the MISEV2014 guidelines. J Extracell Vesicles. 2018 Nov 23;7(1):1535750. doi: 10.1080/20013078.2018.1535750. PMID: 30637094; PMCID: PMC6322352). However, the authors indicate in their work that they specifically reviewed exosome therapies. Why have they focused on that type of extracellular vesicle?

Author Response

The authors carry out this review work that aims to discuss the progress of research based on MSCs/Exo-MSCs to treat CVD or CNSD, explore the potential mechanisms and as a tool in nanomedicine. 

There is quite a bit of controversy about the nomenclature of extracellular vesicles depending on their formation, size, molecules they contain, etc. (Minimal information for studies of extracellular vesicles 2018 (MISEV2018): a position statement of the International Society for Extracellular Vesicles and update of the MISEV2014 guidelines. J Extracell Vesicles. 2018 Nov 23;7(1):1535750. doi: 10.1080/20013078. 2018.1535750. PMID: 30637094; PMCID: PMC6322352). However, the authors indicate in their work that they specifically reviewed exosome therapies. Why have they focused on that type of extracellular vesicle?

Response: Thanks a for the reviewer's constructive comments and suggestion. Several studies have reported that mesenchymal stem cells (MSCs)-derived exosomes have functions similar to those of MSCs, such as repairing tissue damage, suppressing inflammatory responses, and modulating the immune system. Compared with cells, exosomes are more stable and reservable, have no risk of aneuploidy, a lower possibility of immune rejection following in vivo allogeneic administration, and may provide an alternative therapy for various diseases.

This manuscript is a resubmission of an earlier submission. The following is a list of the peer review reports and author responses from that submission.